# Synthesis and Anti-Inflammatory Activity of *N*(2)-Arylindazol-3(2*H*)-One Derivatives: Copper-Promoted Direct *N*-Arylation via Chan–Evans–Lam Coupling

**DOI:** 10.3390/molecules28186706

**Published:** 2023-09-20

**Authors:** Kyungmin Kim, Jeong Ho Kim, Heejae Choi, Byeongno Lee, Jihyun Lee, Kang Min Ok, Tae Hoon Lee, Hakwon Kim

**Affiliations:** 1Department of Applied Chemistry, Global Center for Pharmaceutical Ingredient Materials, Kyung Hee University, Yongin-si 17104, Gyeonggi, Republic of Korea; sp10101@naver.com (K.K.); jeongho1333@gmail.com (J.H.K.); chlgmlwo96@naver.com (H.C.); bnlee@sogang.ac.kr (B.L.); thlee@khu.ac.kr (T.H.L.); 2Department of Chemistry, Sogang University, Seoul 04107, Republic of Korea; jh_christina@naver.com (J.L.); kmok@sogang.ac.kr (K.M.O.)

**Keywords:** anti-inflammatory, cytotoxicity, indazol-3-ones, *N*-arylation, Chan–Evans–Lam coupling

## Abstract

Inflammatory-related diseases are becoming increasingly prevalent, leading to a growing focus on the development of anti-inflammatory agents, with a particular emphasis on creating novel structural compounds. In this study, we present a highly efficient synthetic method for direct *N*-arylation to produce a variety of *N*(2)-arylindazol-3(2*H*)-ones **3**, which exhibit anti-inflammatory activity. The Chan–Evans–Lam (CEL) coupling of *N*(1)-benzyl-indazol-3-(2*H*)-ones **1** with arylboronic acids **2** in the presence of a copper complex provided the corresponding *N*(2)-arylindazol-3(2*H*)-ones **3** in good-to-excellent yields, as identified with NMR, MS, and X-ray crystallography techniques. The cell viability and anti-inflammatory effects of the synthesized compounds (**3** and **5**) were briefly assessed using the MTT method and Griess assay. Among them, compounds **5** exhibited significant anti-inflammatory effects with negligible cell toxicity.

## 1. Introduction

The immune system naturally responds to harmful stimuli such as pathogens, damaged cells, and irritants by triggering inflammation, which is a necessary process for protecting the body. Acute inflammation, a short-term response usually lasting for a few days, aims to remove the source of injury or infection. Chronic inflammation can lead to various complications such as tissue destruction; organ dysfunction; and chronic diseases like arthritis, asthma, atherosclerosis, and cancer. In the pharmaceutical industry, the exploration of novel backbone structures for the development of potential anti-inflammatory drug candidates represents a significant area of research. Although NSAIDs (nonsteroidal anti-inflammatory drugs) and corticosteroids are extensively utilized, there is a demand for more efficient and safer medications to treat chronic inflammation. Emerging strategies in drug development, such as targeted molecule interventions and immune system modulation, hold potential for the advancement of anti-inflammatory drug therapy [1,2,3,4,5].

Indazol-3-ones, depicted in Figure 1, have served as crucial components in various drugs, playing a central role. They have found application in diverse therapeutic areas, including anti-inflammatory agents, anti-hyperlipidemia agents, anti-tumor agents, anti-diabetic agents, and Transient Receptor Potential cation subfamily V member 1 (TRPV1) receptor antagonists [6,7,8,9,10]. The significance of indazol-3-ones as pharmacophores has heightened interest in the development of practical and convenient synthetic methods for generating *N*-substituted indazol-3-one derivatives [11,12,13,14,15], which is crucial because the introduction of different substituents at the *N*-position can significantly affect their biological properties. Therefore, the existence of efficient synthetic methods to prepare *N*-substituted indazol-3-ones is crucial in the discovery of new drug candidates.

In recent years, several synthetic strategies have been developed for the preparation of indazol-3-ones, including *N*-substituted derivatives. To date, several methods to produce *N*(2)-aryl-substituted indazol-3(2*H*)-ones have been reported (Figure 1A), including (i) intramolecular oxidative coupling of *N*-substituted amide and amine using PIFA [16] or IBX [17], (ii) reductive cyclization of *N*-substituted amide with Ti(IV)/Fe catalyst [18] or Zn(II) compound [19], (iii) basic hydrolysis of *N*(2)-substituted pyrazole at high reaction temperature [10], and (iv) the Davis–Beirut reaction of 2-nitrobenzyl alcohol [20]. However, these methods are limited in substrate scope and/or availability of ring cyclization. Hence, there is a need for alternative and convenient synthetic methods for the construction of *N*-aryl-substituted indazol-3-ones.

In this study, a strategy for synthesizing *N*(2)-aryl-substituted indazol-3(2*H*)-ones through *N*(2)-H activation with a commercially or readily available indazol-3(2*H*)-one was considered. Introduction of an aryl group onto nitrogen as a N–H activation reaction is known as the Chan–Evans–Lam (CEL) coupling, in which the nitrogen reacts with a boronic acid in the presence of a copper complex [21,22]. To the best of our knowledge, there have been no reports on the direct arylation of indazol-3-one at the *N*(2) position using CEL coupling, which involves an amide-type nitrogen. Here, we report a highly efficient synthesis of a series of *N*(2)-aryl-substituted indazol-3(2*H*)-ones **3** via an optimized Cu-promoted CEL reaction of *N*(1)-protected indazol-3-one **1** (Figure 1B), followed by deprotection to obtain *N*(1)-H-*N*(2)-aryl-substituted indazol-3-(2*H*)-ones **5**. Finally, the cell viability and anti-inflammatory effects of the synthesized indazol-3-one derivatives (**3** and **5**) were assessed using the MTT method and Griess assay on murine macrophage RAW264.7 cells.

## 2. Results and Discussion

### 2.1. Chemistry

In order to find an efficient synthesis of *N*(2)-aryl-substituted indazol-3(2*H*)-one **3**, the CEL coupling of *N*(1)-protected indazol-3(2*H*)-one **1** with arylboronic acid **2** was investigated based on procedures in the literature to determine optimal conditions [23]. It appears from the results that the type of solvent did not have a significant effect on the reaction yield (Table 1, entries 1–3). However, the amount of copper in the reaction had a significant impact on the yield, with even small increases in copper resulting in higher yields (Table 1, entries 3–6). Replacing the pyridine base with trimethylamine (Et_3_N) resulted in a decrease in yield (Table 1, entries 6–7). Additionally, other types of copper complexes did not react at all under the same conditions (Table 1, entries 9–11). When the reaction was conducted in an argon or oxygen atmosphere, the yield decreased by a small margin (Table 1, entries 12–13). Therefore, the optimized reaction conditions involved using *N*(1)-benzyl-1*H*-indazol-3(2*H*)-one (**1**), *p*-tolylboronic acid (**2a**, 1.5 eq.) and pyridine (20 eq.) in the presence of Cu(OAc)_2_ (0.5 eq.) and CH_2_Cl_2_ solvent in air at room temperature, which resulted in a 91% yield of **3a** (Table 1, entry 8).

The structure of compound **3a**, *N*(2)-aryl-substituted indazol-3(2*H*)-one, was determined by NMR, MS, and single-crystal X-ray diffraction (CCDC 2128503) (Figure 2) (see details in the Appendix A).

To broaden the applicability of the established CEL coupling, we investigated reactions of **1** and various aryl boronic acids **2** (Figure 2, **3a**–**3o**, **4j**–**4m**). Boronic acids possessing electron-donating groups (EDG; alkyl or alkoxy), halogens, or hydrogen on the benzene ring provided the corresponding *N*(2)-aryl-substituted-*N*(1)-benzyl-1*H*-indazol-3(2*H*)-ones **3a**–**3i** in good-to-excellent yields (76–92%). On the other hand, boronic acids **2j**–**2m** containing electron-withdrawing groups (EWG) such as NO_2_, acetyl, nitrile, or CF_3_ at the para-position of the benzene ring gave a regioisomeric mixture of *N*(2)-arylindazol-3(2*H*)-ones **3j**–**3m** and 3-aryloxyindazoles **4j**–**4m** in high yield (90–95%). Here, *N*(2)-arylindazol-3(2*H*)-ones **3j**–**3m** were obtained from *N*(2)-H bond activation of **1**, while 3-aryloxyindazoles **4j**–**4m** were formed via O-H bond activation of the **1** tautomer. CEL coupling reactions with some *N*-heterocyclic boronic acids were also performed to synthesize compounds **3n**–**3o**. Pyridine- or pyrimidine-substituted boronic acids, **2n** or **2o**, produced *N*(2)-substituted indazol-3(2*H*)-ones **3n** or **3o** in moderate yields (71% and 70%), respectively.

Structural differences between indazol-3-ones **3** and 3-aryloxyindazoles **4** were revealed by IR, NMR (^1^H and ^13^C), and MS data of **3l** and **4l**; however, these data could not completely differentiate the two structures. Therefore, to unambiguously distinguish the structures, single crystals of **3l** (CCDC: 2128504) and **4l** (CCDC: 2128505) were grown, and their structures were fully determined with single-crystal X-ray diffraction (Figure 3) (see details in the Appendix A).

To compare the activity of *N*(1)-unsubstituted indazol-3(2*H*)-one **5** with that of *N*(2)-aryl-substituted-*N*(1)-benzyl-1*H*-indazol-3(2*H*)-one **3**, the benzyl group was removed via Pd-catalyzed hydrogenolysis (Figure 3, Method A) [24]. While this reaction proceeded well for most *N*(2)-benzene ring substituents, the yield of some compounds, particularly those with olefin, halogen, acyl, or nitro substituents, decreased. Low yields and poor reproducibility were observed for some compounds, including the nitrile group-substituted derivative **5l** (14%) due to the occurrence of side reactions. Attempts to improve the deprotection using various reaction conditions were unsuccessful in obtaining more effective and reproducible results [25]. Additionally, steric hindrance from a methyl group at the ortho-position resulted in a low yield for **5c** (38%). Therefore, we decided to develop an alternate synthetic route to produce *N*(1)-unsubstituted derivatives **5** for better efficiency and reproducibility.

Since there appeared to be a limitation in using palladium to deprotect compounds with certain substituents, it was necessary to consider the introduction of different protecting groups that utilize non-metallic deprotection conditions. Finally, we chose the *p*-methoxybenzyl (PMB) protecting group as an alternative to the benzyl group and investigated whether the desired compound **5** could be obtained in high yield even with problematic substrates. The PMB group was introduced to indazol-3-one to produce *N*(1)-PMB indazolone **6** according to known methods, and compound **7** was synthesized using CEL coupling (55–91%). Well-known deprotection of the PMB group using TFA (trifluoroacetic acid) was adopted. *N*(1)-PMB-*N*(2)-aryl-substituted indazol-3-one **7** underwent successful deprotection in just one hour under TFA conditions (Figure 3, Method B) [26]. Most of the derivatives, including those that had previously generated low yields with the benzyl-protecting group, were synthesized in very high yields using this method.

### 2.2. Biological Activity

All synthesized indazol-3-ones, **3** and **5**, were briefly screened for cytotoxicity to RAW264.7 cells at a fixed concentration at 20 µg mL^−1^. Compounds **3**, which have substituents at both the *N*(1) and *N*(2) positions, showed cytotoxicity. On the other hand, compounds **5**, which bear a substitution solely at the *N*(2) position, showed acceptable cell viabilities, demonstrated by Formazan formation in the MTT {3-(4,5-dimetnythiazol-2-yl)-2,5-diphenyl-thetazolium bromide} assay (Table 2). As shown in Table 2, cytotoxicity resulted from compounds **3** in which a benzyl group was substituted at the *N*(1) position. From these data, we presume that *N*(1)-substitution affects cell survival.

Anti-inflammatory activities with RAW264.7 cells were then tested with the aforementioned derivatives **5** at five concentrations (0, 1, 5, 10, 20 µg mL^−1^ each), 24 h after inducing inflammation by 1 µg mL^−1^ of LPS (lipopolysaccharides). The nitric oxide concentration, which is correlated with the inflammation level, was determined by the Griess method [27]. Measurements were repeated three times. The mean and standard deviation of the NO concentration (µg mL^−1^) for cells treated by each compound at each concentration are listed in Table 3. As shown in Table 2 and Table 3, of the synthesized compounds, *N*(1)-H indazol-3-ones **5** did not affect cell survival and inhibited the production of NO in LPS-induced RAW264.7 cells. In addition, the difference in activity based on the type of *N*(2) substituent did not follow a clear causal relationship, suggesting that H at *N*(1) is an important feature of the indazol-3-one derivative NO inhibitory activity in LPS-induced RAW264.7 cells. Additional research is underway to further elucidate the relationships between structure and cell toxicity, as well as anti-inflammatory activity.

## 3. Materials and Methods

### 3.1. General Methods

All chemical reagents were purchased from Sigma-Aldrich (St. Louis, MO, USA), Tokyo Chemical Industry (Tokyo, Japan), Alfa Aesar (Morecambe, UK), and Acros Organics (Brookline, MA, USA), and were used without further purification. All glassware was thoroughly dried in a convection oven. Reactions were monitored using thin-layer chromatography (TLC). Commercial TLC plates (silica gel 60 F_254_, Merck Co., Rahway, NJ, USA) were developed and the spots were visualized under UV light at 254 or 365 nm. Silica gel column chromatography was performed with silica gel 60 (particle size 0.040–0.063 mm, Merck Co., Rahway, NJ, USA). Extra-pure-grade solvents for column chromatography were purchased through Samchun Chemicals (Seoul, Republic of Korea) and Duksan Chemicals (Incheon, Republic of Korea). ^1^H and ^13^C NMR spectra were collected with a JEOL ECX-400 spectrometer (at 300 MHz for ^1^H NMR and 75 MHz for ^13^C NMR, Tokyo, Japan) and a JEOL JNM-ECZ400S (at 400 MHz for ^1^H NMR and 100 MHz for ^13^C NMR; Tokyo, Japan). ^1^H NMR spectra chemical shifts are expressed in parts per million (ppm) downfield from tetramethylsilane [Si(CH_3_)_4_], and coupling constants are reported in Hertz (Hz). Splitting patterns are indicated as follows: s, singlet; d, doublet; t, triplet; and m, multiplet. ^13^C NMR spectra are reported in ppm, referenced to chloroform-*d* and DMSO-*d_6_*. Melting points (m.p.) were determined on a Barnstead Electrothermal 9100 instrument (Essex, UK) and were uncorrected. High-resolution mass spectrometry (HRMS) spectra were recorded using a Jeol (JMS-700; Tokyo, Japan).

### 3.2. Chemistry

#### 3.2.1. General Procedure for the Synthesis of Compounds (**3a**–**3o**, **4j**–**4m**)



1-Benzyl-1*H*-indazol-3(2*H*)-one (**1**, 0.36 mmol), Cu(OAc)_2_ (0.18 mmol) and arylboronic acid (0.54 mmol) were dissolved in CH_2_Cl_2_ (0.7 mL) and pyridine (7.2 mmol). The mixture was stirred under air at ambient temperature for 24 h. Upon completion of the reaction, the mixture was quenched with 1 M HCl at 0 °C, followed by extraction with ethyl acetate. The organic layer was separated, washed with brine, and dried over Na_2_SO_4_. After filtration, the mixture was concentrated in vacuo and subsequently purified by silica gel flash column chromatography to yield pure solid products (**3a**–**3o**, **4j**–**4m**).

**3a**, Yield = 91%, white solid, 1-benzyl-2-*p*-tolyl-1*H*-indazol-3(2*H*)-one, m.p.: 164–166 °C. IR (cm^−1^): υ_max_ 1665, 1509, 1319, 1011. ^1^H NMR (300 MHz, DMSO-*d*_6_) δ 7.85 (d, *J* = 8.3 Hz, 1H), 7.67 (t, *J* = 8.0 Hz, 1H), 7.63 (d, *J* = 8.1 Hz, 1H), 7.45–7.38 (m, 4H), 7.20 (t, *J* = 7.5 Hz, 1H), 7.17–7.13 (m, 3H), 6.84 (d, *J* = 5.5 Hz, 2H), 4.91 (s, 2H). ^13^C NMR (75 MHz, CDCl_3_) δ 162.36, 149.81, 136.38, 133.45, 132.57, 132.21, 129.81, 128.61, 128.28, 128.21, 124.50, 124.03, 122.77, 119.82, 113.10, 54.59, 21.11. HRMS (EI) calcd for C_21_H_18_N_2_O (M)^+^ 314.1419, found 314.1418. [CAS RN: 1182783-69-2] [28].

**3b**, Yield = 90%, white solid, 1-benzyl-2-*m*-tolyl-1*H*-indazol-3(2*H*)-one, m.p.: 124–125 °C. ^1^H NMR (300 MHz, DMSO-*d*_6_) δ 7.84 (d, *J* = 8.4 Hz, 1H), 7.67 (t, *J* = 8.1 Hz, 1H), 7.62 (d, *J* = 8.3 Hz, 1H), 7.46 (t, *J* = 7.6 Hz, 1H), 7.36–7.33 (m, 2H), 7.19–7.16 (m, 2H), 7.14–7.12 (m, 3H), 6.84 (d, *J* = 7.3 Hz, 2H), 4.90 (s, 2H), 2.41 (s, 3H). ^13^C NMR (75 MHz, CDCl_3_) δ 162.42, 149.92, 139.16, 135.09, 133.43, 132.29, 128.99, 128.65, 128.27, 128.22, 127.28, 124.62, 124.53, 122.82, 121.09, 119.84, 113.15. 54.78, 21.53. HRMS (EI) calcd for C_21_H_18_N_2_O (M)^+^ 314.1419, found 314.1417. [CAS RN: 2906200-90-0] [28].

**3c**, Yield = 76%, white solid, 1-benzyl-2-*o*-tolyl-1*H*-indazol-3(2*H*)-one, m.p.: 112–113 °C. ^1^H NMR (300 MHz, DMSO-*d*_6_) δ 7.76 (t, *J* = 8.3 Hz, 1H), 7.73 (d, *J* = 7.7 Hz, 1H), 7.70 (t, *J* = 7.3 Hz, 1H), 7.38–7.35 (m, 3H), 7.32–7.29 (m, 1H), 7.23 (t, *J* = 7.7 Hz, 1H), 7.20–7.14 (m, 3H), 6.75 (d, *J* = 6.2 Hz, 2H), 5.16 (d, *J* = 15.4 Hz, 1H), 4.59 (d, *J* = 15.8 Hz, 1H), 1.85 (s, 3H). ^13^C NMR (75 MHz, CDCl_3_) δ 162.09, 149.63, 137.03, 134.72, 133.67, 132.34, 131.40, 128.56, 128.52, 128.20, 127.96, 127.47, 126.49, 124.59, 122.19, 118.31, 111.92, 53.58, 17.97. HRMS (EI) calcd for C_21_H_18_N_2_O (M)^+^ 314.1419, found 314.1417.

**3d**, Yield = 91%, white solid, 1-benzyl-2-phenyl-1*H*-indazol-3(2*H*)-one, m.p.: 131–132 °C. ^1^H NMR (300 MHz, DMSO-*d*_6_) δ 7.87 (d, *J* = 8.2 Hz, 1H), 7.69 (t, *J* = 8.2 Hz, 1H), 7.65 (d, *J* = 8.4 Hz, 1H), 7.60–7.55 (m, 4H), 7.39 (t, *J* = 6.8 Hz, 1H), 7.21 (t, *J* = 7.4 Hz, 1H), 7.19–7.13 (m, 3H), 6.84 (d, *J* = 6.9 Hz, 2H), 4.93 (s, 2H). ^13^C NMR (75 MHz, CDCl_3_) δ 162.43, 150.00, 135.21, 133.27, 132.38, 129.19, 128.63, 128.29, 126.33, 124.55, 123.84, 122.91, 119.83, 113.23, 54.82. (EI) calcd for C_20_H_16_N_2_O (M)^+^ 300.1263, found 300.1259. [CAS RN: 1182783-55-6] [29].

**3e**, Yield = 92%, white solid, 1-benzyl-2-(4-methoxyphenyl)-1*H*-indazol-3(2*H*)-one, m.p.: 152–154 °C. ^1^H NMR (300 MHz, DMSO-*d*_6_) δ 7.81 (d, *J* = 8.4 Hz, 1H), 7.65 (t, *J* = 7.1 Hz, 1H), 7.62 (d, *J* = 7.9 Hz, 1H), 7.42 (d, 9.0 Hz, 2H), 7.18 (t, *J* = 9.0 Hz, 1H), 7.15–7.11 (m, 5H), 6.85 (d, *J* = 7.7 Hz, 2H), 4.88 (s, 2H), 3.82 (s, 3H). ^13^C NMR (75 MHz, CDCl_3_) δ 162.38, 158.25, 149.60, 133.60, 132.17, 128.49, 128.31, 128.21, 127.93, 125.97, 124.48, 122.70, 119.63, 114.52, 112.92, 55.53, 54.41. HRMS (EI) calcd for C_21_H_18_N_2_O_2_ (M)^+^ 330.1368, found 330.1367. [CAS RN: 886974-93-2] [29].

**3f**, Yield = 87%, white solid, 1-benzyl-2-(4-*tert*-butylphenyl)-1*H*-indazol-3(2*H*)-one, m.p.: 117–120 °C. ^1^H NMR (300 MHz, DMSO-*d*_6_) δ 7.82 (d, *J* = 8.3 Hz, 1H), 7.66 (t, *J* = 7.1 Hz, 1H), 7.62 (d, *J* = 7.0 Hz, 1H), 7.60 (d, *J* = 8.6 Hz, 1H), 7.48 (d, *J* = 8.6 Hz, 1H), 7.18 (t, *J* = 7.5 Hz, 1H), 7.15–7.13 (m, 3H), 6.86 (d, *J* = 7.7 Hz, 1H), 4.89 (s, 2H), 1.34 (s, 9H). ^13^C NMR (75 MHz, CDCl_3_) δ 162.38, 149.90, 149.49, 133.59, 132.46, 132.19, 128.62, 128.31, 128.19, 126.16, 124.50, 123.64, 122.80, 119.86, 113.16, 54.73, 34.62, 31.35. HRMS (EI) calcd for C_24_H_24_N_2_O (M)^+^ 356.1889, found 356.1889.

**3g**, Yield = 87%, pink solid, 1-benzyl-2-(4-fluorophenyl)-1*H*-indazol-3(2*H*)-one, m.p.: 133–134 °C. ^1^H NMR (300 MHz, DMSO-*d*_6_) δ 7.85 (d, *J* = 8.2 Hz, 2H), 7.69 (t, *J* = 7.5 Hz, 1H), 7.64 (d, *J* = 8.1 Hz, 1H), 7.57–7.52 (m, 2H), 7.45–7.39 (m, 2H), 7.21 (t, *J* = 7.3 Hz, 1H), 7.15–7.13 (m, 3H), 6.82 (d, *J* = 6.8 Hz, 2H), 4.90 (s, 2H). ^13^C NMR (75 MHz, CDCl_3_) δ 162.59, 162.50, 159.24, 150.06, 133.19, 132.57, 131.31, 131.27, 128.57, 128.40, 128.38, 125.80, 125.68, 124.61, 123.06, 119.64, 116.31, 116.00, 113.20, 54.89. HRMS (EI) calcd for C_20_H_15_FN_2_O (M)^+^ 318.1168, found 318.1167. [CAS RN: 2902600-89-7] [28].

**3h**, Yield = 89%, white solid, 1-benzyl-2-(4-chlorophenyl)-1*H*-indazol-3(2*H*)-one, m.p.: 118–122 °C. ^1^H NMR (300 MHz, DMSO-*d*_6_) δ 7.87 (d, *J* = 8.4 Hz, 2H), 7.70 (t, *J* = 8.2 Hz, 1H), 7.66 (d, *J* = 8.1 Hz, 2H), 7.64 (d, *J* = 9.0 Hz, 2H), 7.56 (d, *J* = 8.8 Hz, 2H), 7.21 (t, *J* = 7.4 Hz, 1H), 7.17–7.10 (m, 3H), 6.81 (d, *J* = 7.5 Hz, 2H), 4.92 (s, 2H). ^13^C NMR (75 MHz, CDCl_3_) δ 162.55, 150.24, 133.89, 132.93, 132.68, 131.77, 12937, 128.63, 128.43, 128.36, 124.79, 124.63, 123.19, 119.72, 113.35, 55.09. HRMS (EI) calcd for C_20_H_15_ClN_2_O (M)^+^ 334.0873, found 334.0873. [CAS RN: 1182783-74-9] [28].

**3i**, Yield = 87%, white solid, 1-benzyl-2-(4-bromophenyl)-1*H*-indazol-3(2*H*)-one, m.p.: 114–116 °C. ^1^H NMR (300 MHz, DMSO-*d*_6_) δ 7.88 (d, *J* = 8.4 Hz, 1H), 7.77 (d, *J* = 8.8 Hz, 2H), 7.70 (t, *J* = 7.1 Hz, 1H), 7.64 (d, *J* = 7.9 Hz, 1H), 7.51 (d, *J* = 8.8 Hz, 2H), 7.21 (t, *J* = 7.1 Hz, 1H), 7.16–7.09 (m, 3H), 6.80 (d, *J* = 7.9 Hz, 2H), 4.92 (s, 2H). ^13^C NMR (75 MHz, CDCl_3_) δ 162.52, 150.27, 134.43, 132.89, 132.71, 132.33, 128.64, 128.44, 128.36, 125.04, 124.64, 123.21, 119.75, 119.62, 113.39, 55.13. HRMS (EI) calcd for C_20_H_15_BrN_2_O (M)^+^ 378.0368, found 378.0364. [CAS RN: 1182783-76-1] [29].

**3j**, Yield = 60%, yellow solid, 1-benzyl-2-(4-nitrophenyl)-1*H*-indazol-3(2*H*)-one, m.p.: 166–168 °C. ^1^H NMR (300 MHz, DMSO-*d*_6_) δ 8.45 (d, *J* = 9.2 Hz, 2H), 7.93 (d, *J* = 8.4 Hz, 1H), 7.86 (d, *J* = 9.0 Hz, 2H), 7.76 (t, *J* = 7.3 Hz, 1H), 7.68 (d, *J* = 7.7 Hz, 1H), 7.25 (t, *J* = 7.5 Hz, 1H), 7.17–7.09 (m, 3H), 6.77 (d, *J* = 6.8 Hz, 2H), 4.97 (s, 2H). ^13^C NMR (75 MHz, CDCl_3_) δ 162.99, 151.10, 144.72, 141.11, 133.57, 132.23, 128.73, 128.50, 124.97, 124.93, 123.88, 122.42, 119.64, 115.64, 113.87, 56.15. HRMS (EI) calcd for C_20_H_15_N_3_O_3_ (M)^+^ 345.1113, found 345.1110. [CAS RN: 1182783-81-8] [29].

**4j**, Yield = 35%, white solid, 1-benzyl-3-(4-nitrophenoxy)-1*H*-indazole, m.p.: 118–120 °C. ^1^H NMR (300 MHz, DMSO-*d*_6_) δ 8.27 (d, *J* = 9.2 Hz, 2H), 7.77 (d, *J* = 8.6 Hz, 1H), 7.50 (d, *J* = 8.0 Hz, 1H), 7.47 (t, *J* = 8.6 Hz, 1H), 7.35–7.23 (m, 7H), 7.15 (t, *J* = 7.1 Hz, 1H), 5.60 (s, 2H). ^13^C NMR (75 MHz, CDCl_3_) δ 161.81, 151.00, 143.24, 141.32, 136.46, 128.80, 127.97, 127.80, 127.23, 125.77, 120.97, 119.49, 117.27, 113.71, 109.78, 53.01. HRMS (EI) calcd for C_20_H_15_N_3_O_3_ (M)^+^ 345.1113, found 345.1111.

**3k**, Yield = 66%, white solid, 2-(4-acetylphenyl)-1-benzyl-1*H*-indazol-3(2*H*)-one, m.p.: 130–132 °C. ^1^H NMR (300 MHz, DMSO-*d*_6_) δ 8.18 (d, *J* = 8.6 Hz, 2H), 7.91 (d, *J* = 8.3 Hz, 1H), 7.73 (d, *J* = 8.6 Hz, 2H), 7.72 (t, *J* = 7.3 Hz, 1H), 7.65 (d, *J* = 7.5 Hz, 1H), 7.23 (t, *J* = 7.3 Hz, 1H), 7.16–7.09 (m, 3H), 6.78 (d, *J* = 6.4 Hz, 2H), 4.95 (s, 2H). ^13^C NMR (75 MHz, CDCl_3_) δ 197.06, 162.70, 150.62, 139.51, 134.20, 133.01, 132.61, 129.58, 128.72, 128.52, 128.37, 124.74, 123.44, 122.52, 119.82, 113.63, 55.57, 26.64. HRMS (EI) calcd for C_22_H_18_N_2_O_2_ (M)^+^ 342.1368, found 342.1365.

**4k**, Yield = 24%, white solid, 1-(4-(1-benzyl-1*H*-indazol-3-yloxy)phenyl)ethanone, m.p.: 95–98 °C. ^1^H NMR (300 MHz, DMSO-*d*_6_) δ 7.93 (d, *J* = 8.8 Hz, 2H), 7.70 (d, *J* = 8.8 Hz, 1H), 7.41–7.37 (m, 2H), 7.29–7.14 (m, 7H), 7.06 (t, *J* = 7.5 Hz, 1H), 5.53 (s, 2H), 2.44 (s, 3H). ^13^C NMR (75 MHz, CDCl_3_) δ 196.69, 160.77, 151.79, 141.27, 136.60, 132.38, 130.41, 128.71, 127.82, 127.54, 127.18, 120.61, 119.72, 116.91, 113.85, 109.64, 52.88, 26.45. HRMS (EI) calcd for C_22_H_18_N_2_O_2_ (M)^+^ 342.1368, found 342.1365.

**3l**, Yield = 63%, white solid, 4-(1-benzyl-3-oxo-1*H*-indazol-2(3*H*)-yl)benzonitrile, m.p.: 96–99 °C. IR (cm^−1^): υ_max_ 2221, 1677, 1595, 1495, 1351. ^1^H NMR (300 MHz, DMSO-*d*_6_) δ 8.05 (d, *J* = 8.4 Hz, 2H), 7.91 (d, *J* = 8.2 Hz, 1H), 7.77 (d, *J* = 8.4 Hz, 2H), 7.74 (t, *J* = 8.0 Hz, 1H), 7.66 (d, *J* = 7.9 Hz, 1H), 7.24 (t, *J* = 7.5 Hz, 1H), 7.19–7.09 (m, 3H), 6.77 (d, *J* = 7.0 Hz, 2H), 4.94 (s, 2H). ^13^C NMR (75 MHz, CDCl_3_) δ 162.85, 150.93, 139.43, 133.36, 133.24, 132.33, 128.68, 128.65, 128.44, 124.83, 123.72, 122.82, 119.63, 118.55, 113.77, 108.94, 55.93. HRMS (EI) calcd for C_21_H_15_N_3_O (M)^+^ 325.1215, found 325.1212. [CAS RN: 1182783-78-3] [29].

**4l**, Yield = 28%, white solid, 4-(1-benzyl-1*H*-indazol-3-yloxy)benzonitrile, m.p.: 79–81 °C. IR (cm^−1^): υ_max_ 2237, 1605, 1481, 1365, 1232, 1167. ^1^H NMR (300 MHz, DMSO-*d*_6_) δ 7.88 (d, *J* = 8.8 Hz, 2H), 7.77 (d, *J* = 8.6 Hz, 1H), 7.50–7.44 (m, 2H), 7.33–7.24 (m, 7H), 7.15 (t, *J* = 7.0 Hz, 1H), 5.60 (s, 2H). ^13^C NMR (75 MHz, CDCl_3_) δ 160.22, 151.01, 141.27, 136.47, 133.97, 128.74, 127.88, 127.70, 127.18, 120.83, 119.49, 118.69, 117.83, 113.72, 109.71, 106.60, 52.93. HRMS (EI) calcd for C_21_H_15_N_3_O (M)^+^ 325.1215, found 325.1212.

**3m**, Yield = 77%, white solid, 1-benzyl-2-(4-(trifluoromethyl)phenyl)-1*H*-indazol-3(2*H*)-one, m.p.: 119–121 °C. ^1^H NMR (300 MHz, DMSO-*d*_6_) δ 7.96 (d, *J* = 8.6 Hz, 2H), 7.91 (d, *J* = 8.3 Hz, 1H), 7.80 (d, *J* = 8.4 Hz, 2H), 7.73 (t, *J* = 8.1 Hz, 1H), 7.66 (d, *J* = 7.9 Hz, 1H), 7.24 (t, *J* = 7.5 Hz, 1H), 7.17–7.12 (m, 3H), 6.79 (d, *J* = 7.5 Hz, 2H), 4.95 (s, 2H). ^13^C NMR (75 MHz, CDCl_3_) δ 162.78, 150.65, 138.52, 133.03, 132.62, 128.69, 128.53, 128.39, 127.93, 127.50, 126.48, 126.46, 126.43, 126.38, 126.33, 125.76, 124.74, 123.46, 122.95, 122.16, 119.73, 113.61, 55.51. HRMS (EI) calcd for C_21_H_15_F_3_N_2_O (M)^+^ 368.1136, found 368.1132.

**4m**, Yield = 14%, colorless oil, 1-benzyl-3-(4-(trifluoromethyl)phenoxy)-1*H*-indazole, ^1^H NMR (300 MHz, DMSO-*d*_6_) δ 7.77–7.74 (m, 3H), 7.47 (d, *J* = 8.3 Hz, 1H), 7.45 (t, *J* = 9.2 Hz, 1H), 7.33–7.27 (m, 5H), 7.23 (d, *J* = 8.0 Hz, 2H), 7.13 (t, *J* = 7.9 Hz, 1H), 5.58 (s, 2H). ^13^C NMR (75 MHz, CDCl_3_) δ 159.46, 151.65, 141.32, 136.64, 128.75, 127.85, 127.60, 127.21, 127.01, 126.99, 126.96, 126.93, 126.91, 125.92, 125.60, 125.16, 122.34, 120.64, 119.75, 117.38, 113.84, 109.66, 52.92. HRMS (EI) calcd for C_21_H_15_F_3_N_2_O (M)^+^ 368.1136, found 368.1136.

**3n**, Yield = 71%, white solid, 1-benzyl-2-(pyridin-3-yl)-1*H*-indazol-3(2*H*)-one, m.p.: 153–156 °C. ^1^H NMR (300 MHz, DMSO-*d*_6_) δ 8.76 (s, 1H), 8.56 (d, *J* = 4.8 Hz, 1H), 7.94 (d, *J* = 6.8 Hz, 1H), 7.89 (d, *J* = 8.1 Hz, 1H), 7.72 (t, *J* = 7.1 Hz, 1H), 7.67 (d, *J* = 7.9 Hz, 1H), 7.62 (dd, *J* = 8.2, 4.8 Hz, 1H), 7.23 (t, *J* = 7.3 Hz, 1H), 7.17–7.10 (m, 3H), 6.81 (d, *J* = 7.7 Hz, 2H), 4.94 (s, 2H). ^13^C NMR (75 MHz, CDCl_3_) δ 162.88, 150.68, 147.05, 144.28, 133.00, 132.62, 132.33, 130.56, 128.58, 128.53, 128.41, 124.66, 123.75, 123.39, 119.49, 113.54, 55.36. HRMS (EI) calcd for C_19_H_15_N_3_O (M)^+^ 301.1215, found 301.1214.

**3o**, Yield = 70%, white solid, 1-benzyl-2-(pyrimidin-5-yl)-1*H*-indazol-3(*2H*)-one, m.p.: 213–216 °C. ^1^H NMR (300 MHz, DMSO-*d*_6_) δ 9.16 (s, 1H), 8.99 (s, 1H), 7.90 (d, *J* = 8.0 Hz, 1H), 7.75 (t, *J* = 7.1 Hz, 1H), 7.69 (d, *J* = 7.9 Hz, 1H), 7.26 (t, *J* = 7.1 Hz, 1H), 7.20–7.10 (m, 3H), 6.86 (d, *J* = 7.7 Hz, 2H), 4.97 (s, 2H). ^13^C NMR (75 MHz, CDCl_3_) δ 162.43, 150.00, 135.21, 133.27, 132.38, 129.19, 128.63, 128.29, 126.33, 124.55, 123.84, 122.91, 119.83, 113.23, 54.82. HRMS (EI) calcd for C_18_H_14_N_4_O (M)^+^ 302.1168, found 302.1168.

#### 3.2.2. General Procedure for the Synthesis of Compounds (**5a**–**5f**, **5l**–**5n**)



*N*(2)-aryl-substituted-*N*(1)-benzyl-1*H*-indazol-3(3*H*)-one (**3**, 0.16 mmol), 20% Pd(OH)_2_/C (0.08 mmol) were dissolved in MeOH (6.4 mL). The mixture was stirred under a hydrogen gas balloon at room temperature for 4 h. The progress of the reaction was monitored using TLC chromatography. Upon completion, the mixture was filtered through celite and washed with CH_2_Cl_2_. The volatile components were evaporated, and the resulting residues subsequently purified through silica gel flash column chromatography to yield pure solid products (**5a**–**5f**, **5l**–**5n**).

**5a**, Yield = 92%, white solid, 2-*p*-tolyl-1*H*-indazol-3(2*H*)-one, m.p.: 209–211 °C. ^1^H NMR (300 MHz, DMSO-*d*_6_) δ 10.64 (s, 1H), 7.80 (d, *J* = 8.4 Hz, 2H), 7.73 (d, *J* = 7.9 Hz, 1H), 7.59 (t, *J* = 7.7 Hz, 1H), 7.36–7.29 (m, 3H), 7.18 (t, *J* = 7.5 Hz, 2H), 2.33 (s, 3H). ^13^C NMR (75 MHz, DMSO-*d*_6_) δ 159.96, 146.41, 135.19, 134.11, 132.32, 129.41, 123.32, 121.75, 118.91, 118.13, 112.55, 20.49. HRMS (EI) calcd for C_14_H_12_N_2_O (M)^+^ 224.0950, found 224.0949. [CAS RN: 74152-88-8] [30].

**5b**, Yield = 83%, white solid, 2-*m*-tolyl-1*H*-indazol-3(2*H*)-one, m.p.: 192–195 °C. ^1^H NMR (400 MHz, DMSO-*d*_6_) δ 10.61 (s, 1H), 7.75–7.73 (m, 3H), 7.60 (t, *J* = 7.8 Hz, 1H), 7.38 (t, *J* = 7.8 Hz, 1H), 7.35 (d, *J* = 8.4 Hz, 1H), 7.19 (t, *J* = 7.4 Hz, 1H), 7.07 (d, *J* = 8.0 Hz, 1H), 2.38 (s, 3H). ^13^C NMR (100 MHz, DMSO-*d*_6_) δ 160.15, 146.53, 138.43, 137.55, 132.49, 128.92, 125.57, 123.41, 121.83, 119.33, 118.10, 116.13, 112.61, 21.24. HRMS (EI) calcd for C_14_H_12_N_2_O (M)^+^ 224.0950, found 224.0948.

**5c**, Yield = 38%, white solid, 2-*o*-tolyl-1*H*-indazol-3(2*H*)-one, m.p.: 142–144 °C. ^1^H NMR (300 MHz, DMSO-*d*_6_) δ 10.64 (s, 1H), 7.80 (d, *J* = 8.4 Hz, 2H), 7.73 (d, *J* = 7.9 Hz, 1H), 7.59 (t, *J* = 7.7 Hz, 1H), 7.36–7.29 (m, 3H), 7.18 (t, *J* = 7.5 Hz, 2H), 2.33 (s, 3H). ^13^C NMR (75 MHz, DMSO-*d*_6_) δ 159.96, 146.41, 135.19, 134.11, 132.32, 129.41, 123.32, 121.75, 118.91, 118.13, 112.55, 20.49. HRMS (EI) calcd for C_14_H_12_N_2_O (M)^+^ 224.0950, found 224.0950. [CAS RN: 74152-87-7] [14].

**5d**, Yield = 92%, white solid, 2-phenyl-1*H*-indazol-3(2*H*)-one, m.p.: 202–205 °C. ^1^H NMR (300 MHz, DMSO-*d*_6_) δ 10.66 (s, 1H), 7.93 (d, *J* = 8.6 Hz, 2H), 7.75 (d, *J* = 7.7 Hz, 1H), 7.61 (t, *J* = 7.6 Hz, 1H), 7.51 (t, *J* = 8.0 Hz, 2H), 7.37 (d, *J* = 8.3 Hz, 1H), 7.22 (p, *J* = 7.3 Hz, 2H). ^13^C NMR (75 MHz, DMSO-*d*_6_) δ 160.20, 146.58, 137.57, 132.52, 129.05, 124.85, 123.41, 121.85, 118.86, 118.08, 112.63. HRMS (EI) calcd for C_13_H_10_N_2_O (M)^+^ 210.0793, found 210.0792. [CAS RN: 17049-65-9] [14].

**5e**, Yield = 86%, white solid, 2-(4-methoxyphenyl)-1*H*-indazol-3(2*H*)-one, m.p.: 177–178 °C. ^1^H NMR (300 MHz, DMSO-*d*_6_) δ 10.62 (s, 1H), 7.79 (d, *J* = 9.0 Hz, 2H), 7.71 (d, *J* = 7.7 Hz, 1H), 7.57 (t, *J* = 7.4 Hz, 1H), 7.33 (d, *J* = 8.1 Hz, 1H), 7.16 (t, *J* = 7.4 Hz, 1H), 7.06 (d, *J* = 9.0 Hz, 2H), 3.78 (s, 3H). ^13^C NMR (75 MHz, DMSO-*d*_6_) δ 159.73, 156.59, 146.26, 132.12, 130.74, 123.25, 121.69, 121.02, 118.08, 114.20, 112.50, 55.33. HRMS (EI) calcd for C_14_H_12_N_2_O_2_ (M)^+^ 240.0899, found 240.0897. [CAS RN: 74152-89-9] [14].

**5f**, Yield = 98%, white solid, 2-(4-*tert*-butylphenyl)-1*H*-indazol-3(2*H*)-one, m.p.: 218–220 °C. ^1^H NMR (400 MHz, DMSO-*d*_6_) δ 10.59 (s, 1H), 7.82 (d, *J* = 7.0 Hz, 2H), 7.74 (d, *J* = 8.0 Hz, 1H), 7.60 (t, *J* = 7.6 Hz, 1H), 7.52 (d, *J* = 7.0 Hz, 2H), 7.36 (d, *J* = 8.4 Hz, 1H), 7.19 (t, *J* = 7.6 Hz, 1H), 1.32 (s, 9H). ^13^C NMR (100 MHz, DMSO-*d*_6_) δ 159.98, 147.46, 146.47, 135.08, 132.39, 125.77, 123.39, 121.82, 118.88, 118.14, 112.63, 34.25, 31.14. HRMS (EI) calcd for C_17_H_18_N_2_O (M)^+^ 266.1419, found 266.1418.

**5l**, Yield = 14%, white solid, 4-(3-oxo-1*H*-indazol-2(3*H*)-yl)benzonitrile, m.p.: 152–154 °C. ^1^H NMR (400 MHz, DMSO-*d*_6_) δ 10.78 (s, 1H), 8.14 (d, *J* = 6.8 Hz, 2H), 7.98 (d, *J* = 6.8 Hz, 2H), 7.79 (d, *J* = 7.6 Hz, 1H), 7.66 (t, *J* = 7.2 Hz, 1H), 7.41 (d, *J* = 8.4 Hz, 1H), 7.23 (t, *J* = 6.8 Hz, 1H). ^13^C NMR (100 MHz, DMSO-*d*_6_) δ 161.11, 147.26, 141.10, 133.49, 133.45, 123.74, 122.35, 118.78, 118.23, 117.71, 112.87, 106.36. HRMS (EI) calcd for C_14_H_12_N_2_O_2_ (M)^+^ 235.0746, found 235.0747.

**5m**, Yield = 74%, white solid, 2-(4-(trifluoromethyl)phenyl)-1*H*-indazol-3(2*H*)-one, m.p.: 233–235 °C. ^1^H NMR (400 MHz, DMSO-*d*_6_) δ 10.75 (s, 1H), 8.17 (d, *J* = 8.8 Hz, 2H), 7.89 (d, *J* = 8.8 Hz, 2H), 7.79 (d, *J* = 8.0 Hz, 1H), 7.66 (t, *J* = 7.6 Hz, 1H), 7.41 (d, *J* = 8.4 Hz, 1H), 7.23 (t, *J* = 7.6 Hz, 1H). ^13^C NMR (100 MHz, DMSO-*d*_6_) δ 160.97, 147.15, 140.76, 133.26, 126.41, 126.37, 125.59, 124.69, 124.37, 123.69, 122.89, 122.28, 118.35, 117.85, 112.86. HRMS (EI) calcd for C_14_H_9_N_3_O (M)^+^ 278.0667, found 278.0663. [CAS RN: 889359-36-8] [30].

**5n**, Yield = 67%, white solid, 2-(pyridin-3-yl)-1*H*-indazol-3(2*H*)-one, m.p.: 183–185 °C. ^1^H NMR (400 MHz, DMSO-*d*_6_) δ 10.73 (s, 1H), 9.14 (s, 1H), 8.45 (d, *J* = 4.8 Hz, 1H), 8.31 (d, *J* = 8.4 Hz, 1H), 7.78 (d, *J* = 8.0 Hz, 1H), 7.65 (t, *J* = 8.0 Hz, 1H), 7.55 (dd, *J* = 8.5, 4.8 Hz, 1H), 7.41 (d, *J* = 8.4 Hz, 1H), 7.23 (t, *J* = 8.0 Hz, 1H). ^13^C NMR (100 MHz, DMSO-*d*_6_) δ 160.83, 147.23, 145.65, 139.91, 134.36, 133.03, 125.76, 123.91, 123.58, 122.20, 117.64, 112.86. HRMS (EI) calcd for C_12_H_9_N_3_O (M)^+^ 211.0746, found 211.0743. [CAS RN: 175653-67-5] [31].

#### 3.2.3. Preparation of N(1)-PMB-1*H*-indazol-3(2*H*)-one **6** [32]



1*H*-indazol-3(2*H*)-one (1 mmol) and NaOH (1 mmol) were dissolved in water (1 mL) at 35 °C, followed by addition of 4-methoxybenzyl chloride (PMB-Cl,1 mmol). The resulting mixture was stirred at 70 °C for 3 h. After the reaction was complete, the solution was cooled to room temperature. The mixture was then quenched with water and extracted with dichloromethane. The organic layer was separated, washed with brine, and dried over anhydrous Na_2_SO_4_. After filtration, the mixture was concentrated under vacuum and subsequently purified through silica gel flash column chromatography to obtain a pure solid product **6**.

White solid, 1-(4-methoxybenzyl)-1*H*-indazol-3(2*H*)-one, m.p.: 158–159 °C. ^1^H NMR (400 MHz, DMSO-*d*_6_) δ 10.66 (br s, 1H), 7.60 (d, *J* = 8.4 Hz, 1H), 7.52 (d, *J* = 8.4 Hz, 1H), 7.31 (t, *J* = 8.0 Hz, 1H), 7.16 (d, *J* = 8.8 Hz, 2H), 6.97 (t, *J* = 8.0 Hz, 1H), 6.84 (d, *J* = 8.8 Hz, 2H), 5.27 (s, 2H), 3.69 (s, 3H). ^13^C NMR (100 MHz, CDCl_3_) δ 159.30, 156.94, 141.89, 128.99, 128.77, 128.69, 121.48, 119.64, 114.17, 113.49, 109.35, 55.32, 52.13. [CAS RN: 1029-30-7] [32].

#### 3.2.4. General Procedure for the Synthesis of Compounds (**7c**, **7g**–**7o**, **8j**–**8m**)



*N*(1)-PMB-1*H*-indazol-3(2*H*)-one (**6**, 0.36 mmol), Cu(OAc)_2_ (0.18 mmol) and arylboronic acid (0.54 mmol) were dissolved in CH_2_Cl_2_ (0.7 mL) and pyridine (7.2 mmol). The reaction was conducted at ambient temperature while being exposed to air. After the reaction was complete, the mixture was quenched with 1 M HCl at 0 °C and then extracted with ethyl acetate. The organic layer was separated, washed with brine, and dried over anhydrous Na_2_SO_4_. Following filtration, the mixture was concentrated under vacuum and subsequently purified through silica gel flash column chromatography to yield pure solid products. (**7c**, **7g**–**7o**, **8j**–**8m**).

**7c**, Yield = 82%, white solid, 1-(4-methoxybenzyl)-2-*o*-tolyl-1*H*-indazol-3(2*H*)-one, m.p.: 103–104 °C. ^1^H NMR (400 MHz, DMSO-*d*_6_) δ 7.77 (d, *J* = 8.0 Hz, 1H), 7.72–7.68 (m, 2H), 7.37–7.31 (m, 4H), 7.22 (t, *J* = 8.0 Hz, 1H), 6.74–6.68 (m, 4H), 5.05 (d, *J* = 15.6 Hz, 1H), 4.52 (d, *J* = 15.6 Hz, 1H), 3.65 (s, 3H) 1.92 (s, 3H). ^13^C NMR (100 MHz, CDCl_3_) δ 162.09, 159.46, 149.61, 136.89, 133.77, 132.24, 131.41, 129.37, 128.44, 127.40, 126.59, 126.47, 124.56, 122.14, 118.45, 113.83, 112.08, 55.20, 53.08, 18.10.

**7g**, Yield = 88%, white solid, 2-(4-fluorophenyl)-1-(4-methoxybenzyl)-1*H*-indazol-3(2*H*)-one, m.p.: 110–112 °C. ^1^H NMR (400 MHz, DMSO-*d*_6_) δ 7.88 (d, *J* = 8.2 Hz, 1H), 7.71 (t, *J* = 7.5 Hz, 1H), 7.67–7.64 (m, 3H), 7.60–7.57 (d, *J* = 8.8 Hz, 2H), 7.22 (t, *J* = 7.5 Hz, 1H), 6.73 (d, *J* = 8.7 Hz, 2H), 6.68 (d, *J* = 8.7 Hz, 2H), 4.85 (s, 2H), 3.63 (s, 3H). ^13^C NMR (100 MHz, CDCl_3_) δ 162.65, 162.03, 159.55, 150.03, 132.49, 131.37, 129.97, 125.67, 125.59, 124.98, 124.60, 123.05, 119.84, 116.26, 116.04, 113.65, 113.36, 55.15, 54.35.

**7h**, Yield = 88%, white solid, 2-(4-chlorophenyl)-1-(4-methoxybenzyl)-1*H*-indazol-3(2*H*)-one, m.p.: 129–130 °C. ^1^H NMR (400 MHz, DMSO-*d*_6_) δ 7.86 (d, *J* = 8.5 Hz, 1H), 7.70 (t, *J* = 8.4 Hz, 1H), 7.65 (d, *J* = 7.5 Hz, 1H), 7.59–7.56 (m, 2H), 7.46–7.41 (m, 2H), 7.22 (t, *J* = 8.0 Hz, 1H), 6.75 (d, *J* = 8.4 Hz, 2H), 6.69 (d, *J* = 8.8 Hz, 2H), 4.84 (s, 2H), 3.64 (s, 3H). ^13^C NMR (100 MHz, CDCl_3_) δ 162.63, 159.57, 150.23, 133.99, 132.63, 131.67, 130.06, 129.39, 124.74, 124.64, 123.19, 119.95, 119.94, 113.64, 113.53, 55.15, 54.57.

**7i**, Yield = 90%, white solid, 2-(4-bromophenyl)-1-(4-methoxybenzyl)-1*H*-indazol-3(2*H*)-one, m.p.: 122–123 °C. ^1^H NMR (400 MHz, DMSO-*d*_6_) δ 7.88 (d, *J* = 8.5 Hz, 1H), 7.78 (d, *J* = 8.8 Hz, 2H), 7.71 (t, *J* = 7.4 Hz, 1H), 7.65 (d, *J* = 7.8 Hz, 1H), 7.53 (d, *J* = 8.8 Hz, 2H), 7.22 (t, *J* = 7.4 Hz, 1H), 6.73 (d, *J* = 8.4 Hz, 2H), 6.68 (d, *J* = 8.8 Hz, 2H), 4.85 (s, 2H), 3.63 (s, 3H). ^13^C NMR (100 MHz, CDCl_3_) δ 162.63, 159.57, 150.23, 133.99, 132.63, 131.67, 130.06, 129.39, 124.74, 124.64, 123.19, 119.95, 119.94, 113.64, 113.53, 55.15, 54.57.

**7j**, Yield = 55%, yellow solid, 1-(4-methoxybenzyl)-2-(4-nitrophenyl)-1*H*-indazol-3(2*H*)-one, m.p.: 179–180 °C. ^1^H NMR (400 MHz, DMSO-*d*_6_) δ 8.46 (d, *J* = 8.8 Hz, 2H), 7.93 (d, *J* = 8.5 Hz, 1H), 7.87 (d, *J* = 8.8 Hz, 2H), 7.77 (t, *J* = 7.4 Hz, 1H), 7.69 (d, *J* = 7.8 Hz, 1H), 7.27 (t, *J* = 7.4 Hz, 1H), 6.71–6.65 (m, 4H), 4.91 (s, 2H), 3.63 (s, 3H). ^13^C NMR (100 MHz, CDCl_3_) δ 163.02, 159.76, 151.07, 144.61, 141.19, 133.45, 130.11, 124.95, 124.90, 124.00, 123.80, 122.32, 119.83, 113.97, 113.74, 55.56, 55.14.

**8j**, Yield = 27%, yellow oil, 1-(4-methoxybenzyl)-3-(4-nitrophenoxy)-1*H*-indazole. ^1^H NMR (400 MHz, DMSO-*d*_6_) δ 8.29 (d, *J* = 9.1 Hz, 2H), 7.78 (d, *J* = 8.5 Hz, 1H), 7.49 (d, *J* = 8.5 Hz, 1H), 7.46 (t, *J* = 7.4 Hz, 1H), 7.33 (d, *J* = 9.4 Hz, 2H), 7.24 (d, *J* = 8.5 Hz, 2H), 7.15 (t, *J* = 7.4 Hz, 1H), 6.89 (d, *J* = 8.7 Hz, 2H), 5.52 (s, 2H), 3.71 (s, 3H). ^13^C NMR (100 MHz, CDCl_3_) δ 161.87, 159.33, 150.85, 143.21, 141.15, 128.72, 128.51, 127.71, 125.77, 120.90, 119.45, 117.22, 114.15, 113.70, 109.84, 55.27, 52.58.

**7k**, Yield = 69%, white solid, 2-(4-acetylphenyl)-1-(4-methoxybenzyl)-1*H*-indazol-3(2*H*)-one, m.p.: 132–134 °C. ^1^H NMR (400 MHz, DMSO-*d*_6_) δ 8.19 (d, *J* = 8.8 Hz, 2H), 7.91 (d, *J* = 8.2 Hz, 1H), 7.76–7.71 (m, 3H), 7.66 (d, *J* = 7.8 Hz, 1H), 7.24 (t, *J* = 7.4 Hz, 1H), 6.72–6.66 (m, 4H), 4.89 (s, 2H), 3.63 (s, 3H), 2.65 (s, 3H). ^13^C NMR (100 MHz, CDCl_3_) δ 197.12, 162.80, 159.65, 150.65, 139.63, 134.16, 132.96, 130.13, 129.61, 124.75, 124.47, 123.43, 122.50, 120.04, 113.78, 113.67, 55.15, 55.07, 26.64.

**8k**, Yield = 27%, white solid, 1-(4-(1-(4-methoxybenzyl)-1*H*-indazol-3-yloxy)phenyl)ethanone, m.p.: 100–101 °C. ^1^H NMR (400 MHz, DMSO-*d*_6_) δ 8.00 (d, *J* = 8.7 Hz, 2H), 7.76 (d, *J* = 8.5 Hz, 1H), 7.44 (t, *J* = 7.5 Hz, 1H), 7.43 (d, *J* = 8.5 Hz, 1H), 7.24–7.20 (m, 4H), 7.11 (t, *J* = 7.5 Hz, 1H), 6.88 (d, *J* = 8.5 Hz, 2H), 5.50 (s, 2H), 3.70 (s, 3H), 2.55 (s, 3H). ^13^C NMR (100 MHz, CDCl_3_) δ 196.78, 160.89, 159.28, 151.41, 141.17, 141.16, 132.41, 130.47, 128.72, 127.50, 120.59, 119.78, 116.94, 114.13, 113.91, 109.75, 55.28, 52.52, 26.50.

**7l**, Yield = 67%, white solid, 4-(1-(4-methoxybenzyl)-3-oxo-1*H*-indazol-2(3*H*)-yl)benzonitrile, m.p.: 148–149 °C. ^1^H NMR (400 MHz, DMSO-*d*_6_) δ 8.06 (d, *J* = 8.8 Hz, 2H), 7.91 (d, *J* = 8.2 Hz, 1H), 7.79 (d, *J* = 8.8 Hz, 2H), 7.75 (t, *J* = 8.4 Hz, 1H), 7.67 (d, *J* = 7.5 Hz, 1H), 7.25 (t, *J* = 8.4 Hz, 1H), 6.71–6.65 (m, 4H), 4.88 (s, 2H), 3.63 (s, 3H). ^13^C NMR (100 MHz, CDCl_3_) δ 162.93, 159.72, 150.92, 139.52, 133.28, 133.26, 130.09, 124.83, 124.15, 123.69, 122.76, 119.85, 118.60, 113.90, 113.72, 108.83, 55.38, 55.14.

**8l**, Yield = 32%, white solid, 4-(1-(4-methoxybenzyl)-1*H*-indazol-3-yloxy)benzonitrile, m.p.: 107–108 °C. ^1^H NMR (400 MHz, DMSO-*d*_6_) δ 7.88 (d, *J* = 8.7 Hz, 2H), 7.77 (d, *J* = 8.5 Hz, 1H), 7.47 (d, *J* = 8.5 Hz, 1H), 7.46 (t, *J* = 7.5 Hz, 1H), 7.29 (d, *J* = 8.8 Hz, 2H), 7.23 (d, *J* = 8.8 Hz, 2H), 7.13 (t, *J* = 7.5 Hz, 1H), 6.88 (d, *J* = 8.7 Hz, 2H), 5.51 (s, 2H), 3.71 (s, 3H). ^13^C NMR (100 MHz, CDCl_3_) δ 160.32, 159.28, 150.86, 141.12, 134.02, 128.69, 128.55, 127.65, 120.79, 119.51, 118.74, 117.79, 114.11, 113.73, 109.79, 106.57, 55.26, 52.54.

**7m**, Yield = 80%, white solid, 1-(4-methoxybenzyl)-2-(4-(trifluoromethyl)phenyl)-1*H*-indazol-3(2*H*)-one, m.p.: 103–104 °C. ^1^H NMR (400 MHz, DMSO-*d*_6_) δ 7.97 (d, *J* = 8.7 Hz, 2H), 7.92 (d, *J* = 8.5 Hz, 1H), 7.82 (d, *J* = 8.5 Hz, 2H), 7.74 (t, *J* = 7.5 Hz, 1H), 7.67 (d, *J* = 7.8 Hz, 1H), 7.25 (t, *J* = 7.5 Hz, 1H), 6.73–6.66 (m, 4H), 4.89 (s, 2H), 3.63 (s, 3H). ^13^C NMR (100 MHz, CDCl_3_) δ 162.89, 159.66, 150.68, 138.62, 132.98, 130.12, 127.80, 127.50, 126.67, 126.42, 124.77, 124.48, 123.45, 122.93, 122.67, 119.96, 113.70, 55.15, 55.03.

**8m**, Yield = 16%, colorless oil, 1-(4-methoxybenzyl)-3-(4-(trifluoromethyl)phenoxy)-1*H*-indazole. ^1^H NMR (400 MHz, DMSO-*d*_6_) δ 7.77 (d, *J* = 8.4 Hz, 2H), 7.76 (d, *J* = 8.4 Hz, 1H), 7.47 (d, *J* = 8.0 Hz, 1H), 7.45 (t, *J* = 7.5 Hz, 1H), 7.31 (d, *J* = 8.7 Hz, 2H), 7.23 (d, *J* = 8.7 Hz, 2H), 7.13 (t, *J* = 7.5 Hz, 1H), 6.88 (d, *J* = 8.5 Hz, 2H), 5.50 (s, 2H), 3.70 (s, 3H). ^13^C NMR (100 MHz, CDCl_3_) δ 159.52, 159.28, 151.53, 141.17, 128.71, 127.52, 127.02, 126.98, 125.52, 125.20, 122.80, 120.59, 119.75, 117.36, 114.13, 113.86, 109.74, 55.27, 52.51.

**7n**, Yield = 71%, white solid, 1-(4-methoxybenzyl)-2-(pyridin-3-yl)-1*H*-indazol-3(2*H*)-one, m.p.: 144–145 °C. ^1^H NMR (400 MHz, DMSO-*d*_6_) δ 8.79 (d, *J* = 2.3 Hz, 1H), 8.57 (dd, *J* = 4.8, 1.4 Hz, 1H), 7.97 (dt, *J* = 8.4, 1.9 Hz, 1H), 7.90 (d, *J* = 8.2 Hz, 1H), 7.73 (t, *J* = 7.4 Hz, 1H), 7.68 (d, *J* = 7.8 Hz, 1H), 7.63 (dd, *J* = 8.2, 4.8 Hz, 1H), 7.25 (t, *J* = 7.4 Hz, 1H), 6.75–6.68 (m, 4H), 4.88 (s, 2H), 3.63 (s, 3H). ^13^C NMR (100 MHz, CDCl_3_) δ 13C-NMR (101 MHz, CHLOROFORM-D) δ 162.97, 159.67, 150.78, 146.98, 144.30, 132.93, 132.47, 130.49, 130.00, 124.69, 124.54, 123.75, 123.38, 119.77, 113.75, 113.70, 55.14, 54.94.

**7o**, Yield = 91%, white solid, 1-(4-methoxybenzyl)-2-(pyrimidin-5-yl)-1*H*-indazol-3(2*H*)-one, m.p.: 165–166 °C. ^1^H NMR (400 MHz, DMSO-*d*_6_) δ 9.18 (s, 1H), 9.01 (s, 2H), 7.92 (d, *J* = 8.5 Hz, 1H), 7.76 (t, *J* = 7.8 Hz, 1H), 7.70 (d, *J* = 7.8 Hz, 1H), 7.27 (t, *J* = 7.5 Hz, 1H), 6.78 (d, *J* = 8.5 Hz, 2H), 6.69 (d, *J* = 8.5 Hz, 2H), 4.91 (s, 2H), 3.64 (s, 3H). ^13^C NMR (100 MHz, CDCl_3_) δ 163.24, 159.87, 155.35, 151.64, 150.25, 133.56, 131.54, 130.02, 124.90, 124.03, 123.91, 119.42, 114.04, 113.94, 55.70, 55.19.

#### 3.2.5. General Procedure for the Synthesis of Compounds (**5c**, **5g**–**5o**)



*N*(1)-PMB-protected *N*(2)-aryl-substituted indazol-3-ones (**7**, 0.15 mmol) were dissolved in trifluoroacetic acid (TFA, 2.3 mL). The mixture was stirred at 60 °C for 1 h. After the reaction was complete, the mixture was concentrated under vacuum and subsequently purified through silica gel flash column chromatography to obtain pure solid products. (**5c**, **5g**–**5o**).

**5c**, Yield = 100%, white solid, 2-*o*-tolyl-1*H*-indazol-3(2*H*)-one.

**5g**, Yield = 99%, white solid, 2-(4-fluorophenyl)-1*H*-indazol-3(2*H*)-one, m.p.: 208–210 °C. ^1^H NMR (400 MHz, DMSO-*d*_6_) δ 10.66 (s, 1H), 7.96–7.92 (m, 2H), 7.75 (d, *J* = 7.6 Hz, 1H), 7.62 (t, *J* = 6.8 Hz, 1H), 7.39–7.34 (m, 3H), 7.20 (t, *J* = 7.2 Hz, 1H). ^13^C NMR (100 MHz, DMSO-*d*_6_) δ 159.73, 156.59, 146.26, 132.12, 130.74, 123.25, 121.69, 121.02, 118.08, 114.20, 112.50, 55.33. HRMS (EI) calcd for C_13_H_9_FN_2_O (M)^+^ 228.0699, found 228.0697. [CAS RN: 135066-29-4] [19].

**5h**, Yield = 99%, white solid, 2-(4-chlorophenyl)-1*H*-indazol-3(2*H*)-one, m.p.: 210–212 °C. ^1^H NMR (400 MHz, DMSO-*d*_6_) δ 10.68 (s, 1H), 7.97 (d, *J* = 6.8 Hz, 2H), 7.76 (d, *J* = 7.6 Hz, 1H), 7.64 (t, *J* = 8.0 Hz, 1H), 7.58 (d, *J* = 6.8 Hz, 1H), 7.38 (d, *J* = 8.4 Hz, 1H), 7.22 (t, *J* = 8.0 Hz, 2H). ^13^C NMR (100 MHz, DMSO-*d*_6_) δ 160.42, 146.76, 136.47, 132.82, 129.00, 128.64, 123.50, 122.07, 120.20, 117.94, 112.72. HRMS (EI) calcd for C_13_H_9_ClN_2_O (M)^+^ 244.0403, found 244.0404. [CAS RN: 17049-63-7] [30].

**5i**, Yield = 99%, white solid, 2-(4-bromophenyl)-1*H*-indazol-3(2*H*)-one, m.p.: 217–219 °C. ^1^H NMR (400 MHz, DMSO-*d*_6_) δ 10.67 (s, 1H), 7.91 (d, *J* = 6.8 Hz, 2H), 7.76 (d, *J* = 8.0 Hz, 1H), 7.70 (d, *J* = 7.2 Hz, 2H), 7.63 (t, *J* = 7.2 Hz, 1H), 7.38 (d, *J* = 8.0 Hz, 1H), 7.23 (t, *J* = 7.2 Hz, 1H). ^13^C NMR (100 MHz, DMSO-*d*_6_) δ 160.43, 146.76, 136.88, 132.83, 131.90, 123.49, 122.07, 120.47, 117.94, 116.77, 112.72. HRMS (EI) calcd for C_13_H_9_BrN_2_O (M)^+^ 287.9898, found 287.9896. [CAS RN: 135066-31-8] [19].

**5j**, Yield = 99%, white solid, 2-(4-nitrophenyl)-1*H*-indazol-3(2*H*)-one, m.p.: 243–245 °C. ^1^H NMR (400 MHz, DMSO-*d*_6_) δ 10.85 (s, 1H), 8.41 (d, *J* = 8.8 Hz, 2H), 8.21 (d, *J* = 9.2 Hz, 2H), 7.80 (d, *J* = 7.6 Hz, 1H), 7.68 (t, *J* = 7.6 Hz, 1H), 7.42 (d, *J* = 8.0 Hz, 1H), 7.24 (t, *J* = 8.2 Hz, 1H). ^13^C NMR (100 MHz, DMSO-*d*_6_) δ 161.32, 147.46, 143.07, 142.74, 133.68, 125.10, 123.84, 122.48, 117.95, 117.61, 112.93. HRMS (EI) calcd for C_13_H_9_N_3_O_3_ (M)^+^ 255.0644, found 255.0645. [CAS RN: 120274-01-3] [33].

**5k**, Yield = 96%, white solid, 2-(4-acetylphenyl)-1*H*-indazol-3(2*H*)-one, m.p.: 233–235 °C. ^1^H NMR (400 MHz, DMSO-*d*_6_) δ 10.77 (s, 1H), 8.12–8.08 (m, 4H), 7.78 (d, *J* = 8.0 Hz, 1H), 7.65 (t, *J* = 8.0 Hz, 1H), 7.40 (d, *J* = 8.4 Hz, 1H), 7.22 (t, *J* = 7.2 Hz, 1H), 2.60 (s, 3H). ^13^C NMR (100 MHz, DMSO-*d*_6_) δ 196.67, 160.88, 147.10, 141.24, 133.21, 132.67, 129.57, 123.66, 122.22, 117.87, 117.62, 112.82, 26.60. HRMS (EI) calcd for C_15_H_12_N_2_O_2_ (M)^+^ 252.0899, found 252.0894.

**5l**, Yield = 99%, white solid, 4-(3-oxo-1*H*-indazol-2(3*H*)-yl)benzonitrile.

**5m**, Yield = 98%, white solid, 2-(4-(trifluoromethyl)phenyl)-1*H*-indazol-3(2*H*)-one.

**5n**, Yield = 95%, white solid, 2-(pyridin-3-yl)-1*H*-indazol-3(2*H*)-one.

**5o**, Yield = 100%, white solid, 2-(pyrimidin-5-yl)-1*H*-indazol-3(2*H*)-one, m.p.: 228–230 °C. ^1^H NMR (400 MHz, DMSO-*d*_6_) δ 10.79 (s, 1H), 9.34 (s, 2H), 9.06 (s, 1H), 7.80 (d, *J* = 8.0 Hz, 1H), 7.68 (t, *J* = 7.2 Hz, 1H), 7.44 (d, *J* = 8.4 Hz, 1H), 7.25 (t, *J* = 7.2 Hz, 1H). ^13^C NMR (100 MHz, DMSO-*d*_6_) δ 161.27, 154.03, 147.80, 146.05, 133.48, 133.00, 123.68, 122.46, 117.15, 113.01. HRMS (EI) calcd for C_11_H_8_N_4_O (M)^+^ 212.0698, found 212.0697.

### 3.3. Biological Activity

#### 3.3.1. Cell Viability Assay (MTT Method)

RAW264.7 cells (Murine macrophages) were cultured using Dulbecco’s modified Eagle medium (DMEM) from Welgene (Seoul, Republic of Korea), supplemented with 10% fetal bovine serum (FBS), 2 mM glutamine, and 100 unit/mL of antibiotics from Gibco BRL (Rockville, MD). The cells were incubated at 37 °C in a humidified atmosphere containing 5% (*v*/*v*) air/CO_2_. For the cell viability assay, RAW264.7 cells (5 × 10^3^/well) were seeded into a 96-well cell culture plate and allowed to attach and stabilize for 18 h. After incubation, the growth medium was replaced with a fresh medium alone. The synthetic compounds were then introduced to the cells at a concentration of 20 µg mL^−1^, and the cells were further incubated for 24 h. Subsequently, the sample-treated culture medium was removed and 100 µg mL^−1^ of 3-(4,5-dimetnythiazol-2-yl)-2,5-diphenyl-thetazolium bromide (MTT) was added to each well. After a one-hour incubation, the purple formazan generated as a result of cellular respiration was dissolved in a 200 µL solution of diemthyl sulfoxide (DMSO), and the absorbance at 560 nm was measured using a multi-plate reader. The analyses were repeated three times, and the results were expressed as the means of three independent experiments.

#### 3.3.2. Measurement of Nitric Oxide Concentration (Griess Assay)

RAW264.7 cells were transferred into 3 × 10^5^ cells per well in a 96-cell culture plate and incubated for 24 h in a 5% CO_2_ incubator at 37 °C. After the incubation, four different concentrations (1, 5, 10, and 20 µg mL^−1^) of synthetic compounds were added to the RAW264.7 cells. Simultaneously, 1 µg mL^−1^ of lipopolysaccharides (LPS) (Sigma-Aldrich, St. Louis, MO, USA) was also introduced and the resulting cells were incubated for an additional 24 h. To measure nitric oxide (NO) concentration in the culture medium, the Griess reagent from Sigma-Aldrich was added in an amount equal to 100 µL of the culture solution, following the manufacturer’s instructions. The absorbance was then measured at 540 nm using a multi-plate reader. To determine the NO concentration, a standard curve was constructed for each concentration of sodium nitrite. The analyses were repeated three times, and the results were expressed as the means of three independent experiments.

## 4. Conclusions

In summary, we developed a practical method to prepare various *N*(2)-arylindazol-3(2*H*)-ones **3** in good-to-excellent yields through CEL coupling of *N*(1)-benzyl-1*H*-indazol-3-(2*H*)-ones **1** with arylboronic acids **2** in the presence of a copper complex. Most compounds **1** reacted with **2** to give only *N*-arylated compounds in excellent yields. However, the reaction of **1** with some boronic acids that included an EWG, such as –NO_2_, –CN, or –CF_3_, resulted in two reaction products, the *N*(2)-arylated indazol-3(2*H*)-ones **3** and the O-arylated products, 3-aryloxyindazoles **4**. The structures of these compounds were determined using single-crystal X-ray diffraction. In addition, the benzyl protecting group of the prepared indazol-3-ones **3** could be removed by Pd-catalyzed hydrogenolysis to yield *N*(1)-H-*N*(2)-aryl indazol-3-ones **5**. For compounds that proved difficult to deprotection, utilizing the PMB protecting group instead of a benzyl group allowed easy deprotection with TFA. We found that most newly synthesized compounds **5**, *N*(1)-H-*N*(2)-aryl indazol-3-ones, did not affect cell survival and inhibited the production of NO in LPS-induced RAW264.7 cells. In addition, the difference in activity according to the type of *N*(2)-substituent did not follow a clear causal relationship, which could mean that H at *N*(1) of the indazol-3-ones is important for NO inhibitory activity in LPS-induced RAW264.7 cells. Additional studies are underway to clarify the structure–activity relationships, and these findings will be reported in the near future.

## Data Availability

The data presented in this study are available in this article or Appendix A.

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
