# Peer review of "Synthesis and Anti-Inflammatory Activity of N(2)-Arylindazol-3(2H)-One Derivatives: Copper-Promoted Direct N-Arylation via Chan–Evans–Lam Coupling"

_molecules, 2023, doi:10.3390/molecules28186706_

Round 1

Reviewer 1 Report

The manuscript entitled “Synthesis and anti-inflammatory activity of N(2)-arylindazol-3(2H)-one derivatives: copper-promoted direct N-arylation via Chan-Evans-Lam coupling by Kyungmin Kim  et al. focuses on the method to prepare various N(2)-arylindazol-3(2H)-ones through CEL coupling of N(1)-benzyl-1H-indazol-3-(2H)-ones with arylboronic acids in the presence of a copper complex, and study them for inflammatory action. The manuscript may be of general interest to the researchers of this field, but the manuscript lacks some information that the author should consider and incorporate in the present form of the manuscript. Here are concerns that need to be addressed in the present form of the manuscript.

1.   The abstract should contain background addressed in a broad context and highlight the purpose of the study.

2.     Introduction is poor for a new references. References for 2018-2023 should be added to the introduction, because only 3 of 21 references are over the past five years.

3.    Section "2.2. Biological activity" should begin with the results of the study in text format and their discussion, not with a table.

4.     Tables 1-4 are schemes. Tables 3 and 4 should be combined into one scheme that integrates both methods of obtaining compounds 5. The scheme for the preparation of compounds S1, 7, 8 should be moved to the text of the article.

5.  In the manuscript, the compounds 3a,b,d,e,g,h,I,j,l and 5a,c,d,I,h,n,g,j synthesized by the authors, are known (Advanced Synthesis & Catalysis (2023), 365(3), 388-396; Synthetic Communications (2009), 39(15), 2647-2663; Tetrahedron Letters (1979), (49), 4765-4768; Synthetic Communications (1991), 21(4), 545-8; Organic Letters (2020), 22(16), 6277-6282; Tetrahedron (1987), 43(20), 4621-4624; Journal of Organic Chemistry (2023), 88(9), 5731-5744; Tetrahedron (2019), 75(30), 4005-4009; Journal of the Chemical Society of Pakistan (1995), 17(4), 232-236; Organic & Biomolecular Chemistry (2018), 16(39), 7236-7244; European Patent Organization, EP284174 A1 1988, etc.). However, the authors presented them as new?! The spectral characteristics of compounds 5c,l,m,n are repeated.

6.   There is no reference control for no one famous anti-inflammatory drugs used in the experiment studies.

Author Response

Please refer to attached file(itemized list_reviewer 1_HKim(20230912).

Reviewer 2 Report

Hakwon Kim and colleagues have made a discovery involving a copper-promoted Chan-Evans-Lam coupling that illustrated anti-inflammatory activity. Following a thorough examination of your manuscript, I have identified several noteworthy concerns that require attention to enhance the overall quality and suitability of the paper for potential publication.

1.    What’s the key improvements of the chemical method described in this manuscript compared to other reported Chan-Evans-Lam coupling such as the one described in doi:10.1002/cbdv.202200327.

2.    Why compound 5l showed a lower yield than the hindered compound 5c during hydrogenation?

3.    Details of the in vitro assay such as duplicated numbers, media, conc. etc. should be described in the table legend.

4.    What is the potential target enzyme of the scaffold synthesized in the manuscript? In other words, why authors decided to test anti-inflammatory activity rather than other bioactivities?

5.    Unit for LD50 and IC50 should be listed in the table.

The quality of English language is fine

Author Response

Please refer to attached file (Itemized list_reviewer 2_HKim(20230912))

Round 2

Reviewer 1 Report

The authors significantly improved the quality of the paper according to reviewer's concerns. I suggest the paper can be accepted.

Reviewer 2 Report

I am happy with the revision and please accept it as it is.

ok